# Evaluating the Quality of Real-World Data on Adherence to Oral Endocrine Therapy in Breast Cancer Patients: How Real Is Real-World Data?

**DOI:** 10.3390/cancers17020200

**Published:** 2025-01-09

**Authors:** A Navarro-Sabaté, R Font, JA Espinàs, J Solà, F Martínez-Soler, M Gil-Gil, G Viñas, A Tibau, M Borrell, M Segui, M Margelí, S Servitja, C Perez, M Domenech, M Nava, M Marin, S Gonzalez, JM Borràs

**Affiliations:** 1Fundamental Care and Clinical Nursing Department, Nursing Faculty, University of Barcelona, L’Hospitalet de Llobregat, 08907 Barcelona, Spain; aureanavarro@ub.edu (A.N.-S.); finamartinez@ub.edu (F.M.-S.); 2Catalan Cancer Plan, Department of Health, L’Hospitalet de Llobregat, 08908 Barcelona, Spain; rfont@iconcologia.net (R.F.); ja.espinas@iconcologia.net (J.E.); jsola@iconcologia.net (J.S.); 3Bellvitge Biomedical Research Institute (IDIBELL), L’Hospitalet de Llobregat, 08908 Barcelona, Spain; 4Clinical Sciences Department, University of Barcelona, L’Hospitalet de Llobregat, 08907 Barcelona, Spain; 5Institut Català d’Oncologia (ICO), IDIBELL, L’Hospitalet de Llobregat, 08908 Barcelona, Spain; mgilgil@iconcologia.net; 6Precision Oncology Group (OncoGir-Pro), Institut d’Investigació Biomèdica de Girona (IDIBGI), 17190 Salt, Spain; gvinyes@iconcologia.net; 7Medical Oncology, Catalan Institute of Oncology, Hospital Universitari Dr. Josep Trueta, 17007 Girona, Spain; 8Program on Regulation, Therapeutics, and Law (PORTAL), Division Pharmacoepidemiology and Pharmacoeconomics, Department of Medicine, Brigham and Women’s Hospital, Harvard Medical School, Boston, MA 02120, USA; atibau@santpau.cat; 9Medical Oncology Department, Hospital Vall d’Hebron, 08035 Barcelona, Spain; mborrell@vhio.net; 10Medical Oncology Department, Consorci Corporacio Sanitaria Parc Tauli, Sabadell and Autonomous University of Barcelona (UAB), 08202 Sabadell, Spain; msegui@tauli.cat; 11Medical Oncology Department, Catalan Institut of Oncology (ICO)-Badalona, B-ARGO (Badalona Applied Research Group in Oncology) in CARE Program at IGTP (Health Research Institute Germans Trias i Pujol), Universitat Autònoma de Barcelona, 08916 Badalona, Spain; mmargeli@iconcologia.net; 12Medical Oncology Department, Hospital del Mar, 08019 Barcelona, Spain; sservitja@psmar.cat; 13Medical Oncology Department, Xarxa Sanitària i Social Santa Tecla, 43003 Tarragona, Spain; 14Fundació Althaia, Medical Oncology Department, Hospital de Manresa, 08240 Manresa, Spain; mdomenech@althaia.cat; 15Medical Oncology Department, Hospital de Mataró, 08304 Mataró, Spain; mnava@csdm.cat; 16Medical Oncology Department, Consorci Sanitari de Terrassa, 08227 Terrassa, Spain; mmarin@cst.cat; 17Medical Oncology Department, Hospital Mútua Terrassa, 08221 Terrassa, Spain; soniagonzalez@mutuaterrassa.es

**Keywords:** breast cancer, oral endocrine therapy, real-world-data, adherence validation

## Abstract

The objective of this study was to evaluate the validity of RWD-based adherence measurements obtained from accessible electronic health records in the Spanish national health system in all women diagnosed with breast cancer in the public healthcare system in Catalonia (Spain). Our results showed that nonadherence during the first year of treatment was around 11% in both cohorts, analysed using the RWD, and without significant differences between them. Furthermore, determinants associated with nonadherence (age and type of oral endocrine treatment) were similar in both approaches used. The results also show that it is fast and feasible to use RWD to identify individuals who are not refilling prescriptions as often as they should. In conclusion, the validity of the RWD method to estimate adherence has been confirmed and, at the same time, this method provides valuable evidence to help oncologists discuss adherence with their patients.

## 1. Introduction

In recent years, with the widespread implementation of electronic databases and the digitisation of medical records, there has been a marked increase in the use of real-world data (RWD) generated during healthcare activity to evaluate therapies and care patterns [1].

Among patients who receive oral treatments, such as oral endocrine therapy (OET) for breast cancer, there are specific factors affecting adherence that differ from those associated with clinician-administered treatments. It is possible to measure adherence to oral treatment based on RWD from clinical–administrative prescription refill records of community and hospital pharmacies. Various published studies [2,3,4] demonstrate the availability and international acceptance of this adherence measure [5].

Breast cancer is the most common cancer among women worldwide, with an estimated 2.3 million new cases in 2020 [6]. In Catalonia (Spain), breast cancer accounts for 28.1% of all cancer diagnoses, and its impact is expected to grow in the near future [7]. More than 70% of breast cancers have oestrogen and/or progesterone receptors [8,9]. Endocrine therapy, with tamoxifen and/or aromatase inhibitors, is a mainstay of adjuvant treatment for hormone receptor-positive breast cancer, as reflected in clinical guidelines [10,11,12]. A large body of research has demonstrated the benefits of these drugs in reducing the risk of recurrence and breast cancer-specific mortality [13]. The recommended duration of endocrine therapy is at least five years, although some authors argue that the long-term risk of recurrence warrants a longer regimen, provided the risk–benefit balance is favourable [14,15,16]. In Spain, endocrine therapy is prescribed by hospital specialists and dispensed in community pharmacies.

Multiple studies have identified and recognised therapeutic adherence as a crucial and clinically relevant factor associated with recurrence and survival [2,17,18]. There are several methods for assessing treatment adherence [19]. Indirect methods such as patient interviews are by far the most widely used; however, they can be biassed and generally overestimate adherence [20]. Analysing prescriptions and refills is an increasingly popular alternative method [21,22] owing to the spread of information technologies such as electronic prescriptions.

The objective of this study was to evaluate the validity of RWD-based adherence measurements. If these estimates are accurate, they could help professionals identify nonadherent patients for personalised discussions about the importance of continuing to use OET at the prescribed dose.

## 2. Materials and Methods

We based our analysis on two retrospective cohorts.

### 2.1. Cohort 1: Based on Real-World Data

The first cohort was designed to evaluate adherence to OET among women who started treatment in 2021 after being diagnosed with breast cancer in a public hospital in Catalonia. To obtain data for the analysis, we combined the community pharmacy billing registry, electronic hospital discharge records (Conjunto Mínimo Básico de Datos; CMBD), and the Catalan health division’s central insurance registry. The records came from clinical and administrative databases. We followed this cohort for one year after initiation of therapy.

The pharmacy billing registry is a mandatory record of all prescriptions funded by the Catalan public health system that are dispensed in pharmacies. For each eligible patient, we recorded the date of dispensing, the type of drug administered (tamoxifen, aromatase inhibitors), and the number of pills dispensed. The date of first prescription fill was considered the start date of OET. This cohort was linked to the Catalan health division’s central insurance registry to determine each patient’s vital status and the date of death, if applicable.

The CMBD is a population-based registry that includes information on pathologies treated in public hospitals in Catalonia. It was our main source of information for identifying women with a primary diagnosis of breast cancer first recorded in the hospital discharge report. Patients treated in private centres were excluded, as their data were not available in the CMBD. We also recorded age at diagnosis and information about surgical procedures directly rated to breast cancer. The CMBD does not include cancer stage or hormone receptor status.

Adherence has been defined as “the degree to which use of medication by the patient corresponds with the prescribed regimen” [23]. To estimate adherence, we calculated the proportion of days covered by prescription refills during the treatment period (first year after OET initiation); we considered an accumulated adherence rate of 80% or greater to be satisfactory [24]. Any change between tamoxifen and an aromatase inhibitor was considered treatment continuation (sequential regimen).

With this information, the attending physician could determine which women were nonadherent. By reviewing the medical records of a selection of these women, the attending physician classified reasons for nonadherence as follows: no clear reason (patient decision), adverse effects, change of therapy due to disease progression, change to chemotherapy for other reasons, stage IV cancer, transfer to a private hospital, change of residence to another Spanish autonomous community, and other reasons.

### 2.2. Cohort 2: Breast Cancer Patients Identified in Population-Based Cancer Registries

The second retrospective cohort was from a previous study, published in 2019 [2]. Participants were women diagnosed with breast cancer from 2007 to 2011 and included in the population-based cancer registries of two Catalan provinces (Girona and Tarragona). These cancer registries contain information on sociodemographic variables, tumour characteristics, hormone receptor status, diagnosis and stage, surgical treatment, neoadjuvant treatment, and adjuvant treatment. Font and colleagues had identified breast cancer diagnoses using International Classification of Diseases 10th revision [ICD-10] codes (C50, D05.1, D05.7, or D05.9), and they followed participants for five years. In the present study, we reanalysed the data to estimate the adherence of this cohort at one year of follow-up, reproducing the conditions of the RWD analysis (i.e., considering only age and hormonal treatment). We then compared the adherence estimate with that obtained from cohort 1 to evaluate the validity of measuring adherence with RWD. Finally, we carried out a second reanalysis of cohort 2, incorporating clinical variables to determine their impact on adherence.

### 2.3. Statistical Analysis

For both cohorts, we performed exhaustive data cleaning to obtain high-quality databases; then, we performed a descriptive analysis of the data.

When analysing determinants of adherence in the second cohort, we used unconditional univariate logistic regression to estimate the probability of a patient being adherent. First, we included only the RWD variables (age and type of oral hormonal treatment); then, we performed a second reanalysis with the variables available in the previous study (age, stage, neoadjuvant treatment, surgery, and type of oral hormone treatment (tamoxifen, aromatase inhibitors, and sequential regimen)). We carried out a multivariate logistic regression to adjust for these variables, with adherence as the dependent variable. The results were expressed as odds ratios (ORs) with 95% confidence intervals (CIs). A two-tailed *p* value below 0.05 was considered to indicate statistical significance. All analyses were performed using SPSS software (version 21).

## 3. Results

### 3.1. Analysis of Cohort 1

Figure 1 illustrates the patient selection procedure for the first cohort, combining data from three registries: the community pharmacy billing registry, CMBD, and Catalan health division’s central insurance registry. Of 5915 women who began OET in 2021, 4868 had a breast cancer diagnosis (65,967 refills). Patients with no data in the CMBD, or with a primary diagnosis other than breast cancer, were excluded from the analysis (2048 women).

The first cohort included the 3867 women with BC who began hormone treatment in 2021. Table 1 presents a descriptive analysis of the cohort. It shows that 20.6% of women were aged under 50 years and 71.8% were taking aromatase inhibitors. These variables were associated with each other (*p* < 0.001).

The rate of non-adherence at one year of follow-up based on RWD (available from the pharmacy billing registry and the CMBD) was 10.9%.

Oncologists reviewed the medical records of 286 nonadherent women (67.9% of the total of nonadherent women) to collect the number of OET interruptions and reasons for interruptions. There were treatment interruptions documented in 59.4% of reviewed medical records. Among these records, 64.1% listed only one reason for treatment interruption, 32.4% listed two reasons, and 3.5% listed three reasons. Table 1 shows that 48.8% of women interrupted treatment because of adverse effects (older age was correlated with more interruptions due to adverse effects), 40.0% had no clear reason, 29.4% had other reasons related to breast cancer (advanced stage, progression, negative hormone receptors, treatment changes), 14.7% had other clinical reasons, and 6.5% were lost to follow-up for different reasons (transfer to other Spanish autonomous communities or to a private hospital).

The percentage of nonadherence based on the review of medical records of nonadherent women was 6.9%. It is important to highlight that of the total number of women who were nonadherent prior to the review, 63.6% were confirmed as nonadherent after the review of their medical records due to not having clinical or other reasons for interruption recorded/known to justify the suspension. For the 40.6% of nonadherent women with no mention of treatment interruption in their medical records, the oncologist was alerted following the report made based on RWD.

### 3.2. Reanalysis of Cohort 2

In the second cohort, the overall percentage of nonadherence at one year of treatment was 11.3% from the RWD subpopulation. There was no significant difference between this estimate and the estimate of 10.9% from cohort 1 (*p* = 0.619).

Table 2 presents the results of the two reanalyses: including the variables/information obtained from the review of the medical records (n = 2413) and including only the RWD variables (n = 2992). Women younger than 50 years and women who received tamoxifen or a sequential regimen had lower rates of adherence. Compared with women aged under 50 years, the probability of adherence was higher in women aged 50 to 69 years (RWD variables: OR 1.76, 95% CI 1.35–2.30; medical record variables: OR 1.68, 95% CI 1.24–2.26) and in women older than 69 years (RWD variables: OR 1.71, 95% CI 1.28–2.29; medical record variables: OR 2.12, 95% CI 1.41–3.18). Compared with women who received only tamoxifen, the probability of adherence was higher among those who received aromatase inhibitors (RWD variables: OR 1.94, 95% CI 1.49–2.53; medical record variables: OR 2.21, 95% CI 1.61–3.04). The first and second reanalysis produced similar results for the independent variables of age and type of OET.

## 4. Discussion

The wider use of technology-driven digital support services (e.g., electronic health records) and increased capacity for data storage and data analysis have led to the rapid availability of RWD [1,25]. RWD provide opportunities to identify and learn patterns for clinical prognostication and improve predictions in selected outcomes, especially if linked with administrative data.

In this study, we assessed the validity of RWD for estimating nonadherence to OET in people with breast cancer. Different studies using population-based data from around the world have proven that nonadherence to endocrine therapy is significantly and independently associated with recurrence and all-cause mortality [9,26,27,28,29,30]. Therefore, RWD may be useful for evaluating the use and discontinuation of oral therapies and for investigating possible mitigation strategies.

Our research team recently used RWD to assess the impact of COVID-19 on adherence to oral chemotherapy among people with newly diagnosed breast cancer [3]. In the present study, we used RWD to focus on a nonadherent subpopulation in the first year of treatment and to make clinicians aware of this nonadherent subpopulation so they could contact the relevant patients and discuss the risks of nonadherence.

Our results show that it is easy, fast, and feasible to collect data from hospital discharge records (CMBD) and link them to pharmacy records (two easily accessible RWD sources in the Spanish national health system) to identify individuals who are not refilling prescriptions as often as they should. If health professionals have access to these results, they can contact nonadherent women for personalised follow-up and investigate specific reasons for treatment interruptions. The rate of nonadherence estimated from RWD was higher than the estimate based on a review of medical records by the attending oncologist: 10.9% versus 6.9%. This difference may be explained by the inclusion of all stages of breast cancer. However, this difference does not necessarily limit the validity of the RWD method, which does not underestimate or substantially overestimate the result based on the review of medical records. The fact that a large percentage of nonadherent women (40.6%) had no recorded treatment interruptions in their medical records (which we discovered when we compared the different data sources) highlights the utility of analysing pharmacy dispensing records to identify adherence issues, particularly considering that reports made on RWD can alert the oncologist and identify patients who may be medicating with less than the desired dose without being aware of it.

With the second cohort (taken from a previous study of people diagnosed with breast cancer between 2007 and 2011 in Tarragona and Girona), we performed a multivariable analysis of adherence in the first year of treatment, considering only RWD variables and then using all the cancer registry variables. These two types of analyses produced similar non-adherence estimates (11.3% with RWD variables, 10.5% with all cancer registry variables). Furthermore, a comparison between the first and second cohorts pointed out that there was no significant difference between estimates when comparing the overall percentage of nonadherence at one year of treatment from RWD (10.9% from cohort 1 versus 11.3% from cohort 2).

The benefit of reanalysing previous data with criteria that simulate the RWD-based approach is that we can include all cases from a cancer registry (people treated in public and private hospitals), with verified information on stage and relevant clinical variables. The fact that the determinants associated with nonadherence are similar, even in magnitude, reinforces the conclusions of the RWD analysis.

Participants from the previous study were followed for five years, and the rate of nonadherence at one year (11.3%) increased to 15.5% at five years and was associated with a higher risk of recurrence and cancer death [2]. This highlights the utility of RWD for the early detection of nonadherence after the first year of treatment.

In both analyses, nonadherence was significantly associated with age and type of treatment, in line with previous studies [2,3,28]. Age under 50 years and initial treatment with tamoxifen increased the probability of nonadherence [2,3,28]. Both variables were available in the RWD database used in this study; thus, the approach used here, based on the linkage between population-based cancer registry databases and the reimbursement of refill drugs, would be a feasible method to assess adherence in BC patients with oral endocrine therapy.

Although RWD in Spain do not include structured information on breast cancer staging, we can conclude that the RWD-based method is robust and valid in the context described in this article: as an instrument for the attending physician to evaluate adherence and discuss the clinical implications of treatment interruptions with nonadherent patients. Including this information on adherence from medical records could constitute a fundamental advance in the more precise use of information obtained from clinical databases, helping physicians make better use of their time by identifying the patients who require personalised follow-up. Furthermore, by reviewing this information during follow-up visits, the oncologist can identify patients who are unaware of their nonadherence (unintentional non-adherence) [31,32] and intervene to help them understand the importance of complying with their prescribed medication dose. While the method presented in this article is based on nonadherence during the first year, oral therapy for breast cancer normally lasts five years or more [14,15,16]. A study with a longer follow-up would likely find a difference in unintentional nonadherence over the next four years of treatment. In fact, some studies have reported variations in adherence for every year of treatment [33].

In conclusion, this article describes one method of using RWD from different health databases to create effective strategies aimed at improving adherence. The RWD-based method could be useful in the near future, particularly in evaluating the impact of CDK4/6 inhibitors—an increasingly utilised adjuvant treatment—on adherence to hormonal therapy. It could also be extrapolated to other types of cancer treated with oral therapy.

## 5. Conclusions

This article describes one method of using easily accessible RWD electronic health records in the Spanish national health system, which is robust and valid as an instrument to create effective strategies aimed at improving adherence to the first year of breast cancer oral treatment follow-up. This method could also be potentially extrapolated to other types of cancer treated with oral therapy, providing valuable evidence to help oncologists discuss adherence with their patients.

## Figures and Tables

**Figure 1 cancers-17-00200-f001:**
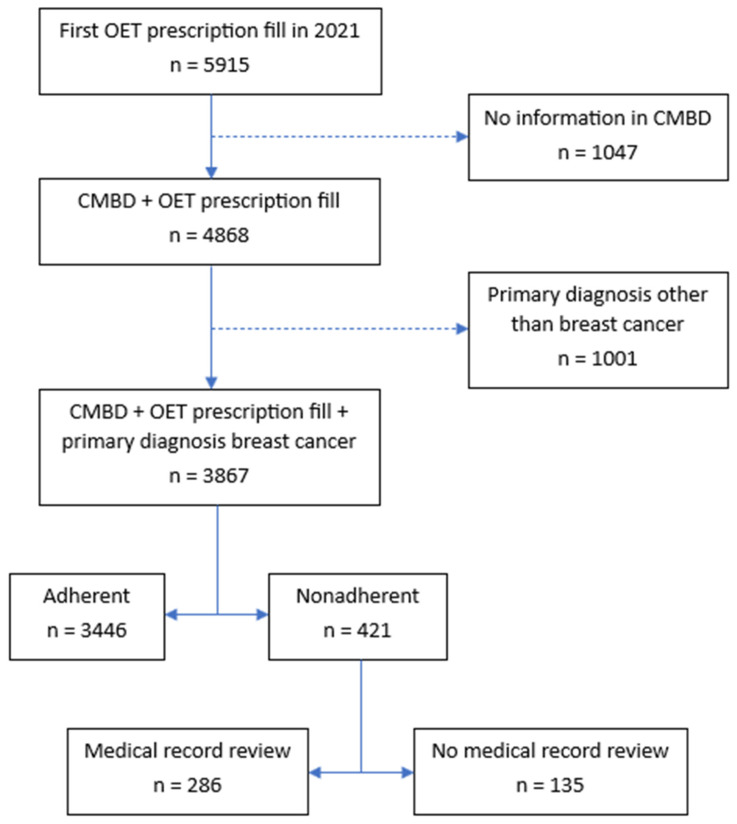
Flow diagram of all patients diagnosed with breast cancer through the public health system of Catalonia (Spain) who first filled a prescription for oral endocrine therapy in a community pharmacy in 2021. Abbreviations: CMBD, hospital discharge records in the national health system (Conjunto Mínimo Básico de Datos); n, number of patients; OET, oral endocrine therapy.

**Table 1 cancers-17-00200-t001:** Description of women in 2021 cohort (at 1 year of follow-up).

	n	%
**Age group**		
<50 years	797	20.6%
50–69 years	1994	51.6%
≥70 years	1076	27.8%
**Type of OET**		
Tamoxifen	875	22.6%
Aromatase inhibitors (AI)	2776	71.8%
Sequential regimen	216	5.6%
Tamoxifen + AI	76	35.2%
AI + tamoxifen	133	61.6%
AI + tamoxifen + AI	7	3.2%
**Adherence to OET**		
Yes	3446	89.1%
No	421	10.9%
**MR review in nonadherent patients**		
Yes	286	67.9%
No	135	32.1%
**Interruptions reported in MR**		
Yes	170	59.4%
No	116	40.6%
**Number of reasons for interruptions**		
1	109	64.1%
2	55	32.4%
3	6	3.5%
**Reasons for interruptions**		
Patient decision	68	40.0%
Adverse effects	83	48.8%
Disease progression	18	10.6%
Negative hormone receptors	10	5.9%
Change to chemotherapy	6	3.5%
Stage IV cancer	16	9.4%
Transfer to private hospital	2	1.2%
Transfer within Catalonia	1	0.6%
Transfer outside Catalonia	8	4.7%
Other	25	14.7%

Abbreviations: MR, medical record; OET, oral endocrine therapy.

**Table 2 cancers-17-00200-t002:** Comparison of adherence in the population cohort (Tarragona and Girona between 2007 and 2011) at one year of follow-up (using only real-world data variables and using all study variables).

	Real-World Data Variables ^a^				Medical Record Review Variables ^b^			
	n	% adh.	aOR (95% CI)	*p* Value	n	% adh.	aOR (95% CI)	*p* Value
**Age group**								
<50 years	794	84.3	1		676	83.7	1	—
50–69 years	1304	90.4	1.76 (1.35–2.30)	<0.001	1122	90.5	1.68 (1.24–2.26)	<0.001
≥70 years	894	90.2	1.71 (1.28–2.29)	<0.001	615	91.2	2.12 (1.41–3.18)	<0.001
**Cancer stage at diagnosis**								
I	—	—	—	—	965	91.2	1	—
II	—	—	—	—	1011	88.7	0.90 (0.63–1.29)	0.562
III	—	—	—	—	437	83.5	0.61 (0.39–0.93)	0.023
**Neoadjuvant treatment**								
Yes	—	—	—	—	407	83.5	1	—
No	—	—	—	—	2006	89.8	1.42 (0.90–2.24)	0.13
**Surgery**								
No	—	—	—	—	31	83.9	1	—
Yes	—	—	—	—	2382	88.8	1.17 (0.37–3.66)	0.791
**Type of treatment**								
OET	—	—	—	—	279	84.9	1	—
CTx-RT-OET	—	—	—	—	998	88.2	1.43 (0.88–2.33)	0.147
RT-OET	—	—	—	—	1136	90.2	1.82 (1.17–2.81)	0.007
**Type of OET**								
Tamoxifen	743	84.5	1		522	83.3	1	—
Aromatase inhibitors	1626	91.4	1.94 (1.49–2.53)	<0.001	1352	91.9	2.21 (1.61–3.04)	<0.001
Sequential regimen	558	86.9	1.22 (0.89–1.67)	0.224	496	86.7	1.27 (0.89–1.82)	0.182
Unknown/other	65	84.6	1.01 (0.50–2.03)	0.984	43	81.4	1.10 (0.47–2.59)	0.828

^a^ Reanalysis 1: whole cohort, univariate analysis (n = 2992; 11.3% nonadherent). ^b^ Reanalysis 2: women with invasive stage I-III hormone receptor-positive breast cancer; multivariable analysis adjusted for age, stage, neoadjuvant treatment, surgery, and adjuvant treatment (n = 2413; 10.5% nonadherent). Abbreviations: adh., adherence; aOR, adjusted odds ratio; CI, confidence interval; CTx, chemotherapy, n, number of patients; OET, oral endocrine therapy, RT, radiotherapy.

## Data Availability

The datasets generated during and analysed during the current study are not publicly available due to confidentiality measures but are available from the corresponding author on reasonable request.

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
