# Peer review of "Evaluating the Quality of Real-World Data on Adherence to Oral Endocrine Therapy in Breast Cancer Patients: How Real Is Real-World Data?"

_cancers, 2025, doi:10.3390/cancers17020200_

Round 1

Reviewer 1 Report

Comments and Suggestions for Authors

The present study was undertaken to evaluate the validity of real-world data-based adherence measurements obtained from accessible electronic health record in the Spanish national health System in all women diagnosed with breast cancer in the public healthcare system in Catalonia. The authors have conducted two retrospective cohort studies. Cohort 1 (RWD) consisted of women diagnosed with breast cancer in 2021 in the public healthcare system of Catalonia. Data for cohort 2 came from two population-based cancer registries in Girona and Tarragona (Catalonia), with diagnoses from 2007 to 2011. The authors concluded that their study confirmed the validity of estimating adherence with RWD from the Spanish national health system, although when combined with reviewing medical records may provide more reliable and higher quality data. Therefore, the RWD method provides valuable evidence to help oncologists discuss adherence with their patients.

The authors have analyzed the various data with well-described statistical methodology and have clearly presented and discussed their findings. Although their data are coming from a small European area, the final observations may help the health systems, together, of course, with similar studies from other regions/countries. Moreover, their method could also be extrapolated to other types of cancer treated with oral therapy.

Reviewer 2 Report

Comments and Suggestions for Authors

Abstract: Need to discuss the 2nd cohort; the second sentence mentioned “both cohorts” but only one cohort is described in the first sentence

In line 176, it is mentioned that the variables are associated with each other. Not sure what this means, associated with each other? In what way?

In lines 180-190, I am confused about the 63.6% vs 40.6% - how were the other 23% of women determined to be non-adherent after review of medical records, if not that there was no documentation of treatment interruption?

In line 199-200, the "RWD subpopulation" is mentioned but what is the RWD subpopulation? I thought the whole cohort was analyzed.

In line 248, you mention it’s fast and easy - how many hours of work and what specific resources (number of people, technology used to create the database?) did it take to create the final cohort 1? Including the details regarding resources and time needed to complete the data gathering would be nice to include in the results if you conclude it was fast and easy.

Can you clarify how a patient would take less than desired dose without being aware of it?

Is there a word missing in line 285?

How do results of your study compare with other studies evaluating non-adherence? Were there differences depending on method of determining adherence? Since you are looking at using RWD, it may be nice to include other papers that have done similar work. Just 2-3 sentences summarizing other papers. This paper below details several relevant studies you may want to look at.

\https://aacrjournals.org/cancerpreventionresearch/article/7/4/378/50346/Adherence-to-Endocrine-Therapy-in-Breast-Cancer

Reviewer 3 Report

Comments and Suggestions for Authors

The Authors performed an interesting analysis on the quality of real-world data on adherence to oral endocrine therapy in breast cancer patients, proposing an accessible alternative. The paper was well written demonstrating a remarkable experience in the issue. The References are appropriate and the quality of English does not limit my understanding of the research.

The article is suitable for the publication.